



# A machine-learning reference dataset for SO₂ plumes observed by TROPOMI: uncertainties and emission estimates

Douglas P. Finch[1,2] and Paul I. Palmer[1,2]

[1]School of GeoSciences, University of Edinburgh, Edinburgh, UK
[2]National Centre for Earth Observation, University of Edinburgh, Edinburgh, UK

**Correspondence:** Douglas Finch (d.finch@ed.ac.uk)

**Abstract.** Sulphur dioxide (SO₂) is a major atmospheric pollutant from fossil fuel combustion, metal smelting, and volcanic degassing, impacting human health, acid deposition, and climate forcing. Existing emission inventories are often temporally lagged and spatially coarse, failing to capture high-intensity, sporadic events. To address this, we present a novel, near real-time approach using a U-Net image segmentation model to automatically isolate SO₂ plumes from over 31,000 TROPOMI satellite
swaths (Jan 2019-–Dec 2024). The model successfully identified 53,993 individual plumes. The highest annual detection rate in 2019 was attributed to massive stratospheric SO₂ injections from the Raikoke and Ulawun volcanic eruptions. Clustering analysis confirmed plume origins around expected volcanic and industrial hotspots (e.g., Iztaccíhuatl, Norilsk), with volcanic sources dominating the top ten clusters. We derived rapid, physics-informed emission rate estimates for each plume, finding a median rate of 14,629 kg hr⁻¹. This detection threshold for this approach, which we estimate to be ~524 kg hr⁻¹, is four orders
of magnitude larger than typical fluxes in the EDGAR inventory, demonstrating the utility of the plume database for detecting extreme, high-intensity events. However, the algorithm struggles to detect sources in high-background regions like China, where high SO₂ saturation likely prevents individual plume isolation. This study demonstrates machine learning as a powerful tool for transforming atmospheric monitoring, providing the high-cadence, fine-grained quantification of SO₂ emissions crucial for validating global inventories and ensuring effective environmental management.

# 1 Introduction

Sulphur dioxide (SO₂) is an atmospheric pollutant predominately produced from fossil fuel combustion for power generation, residential heating, industrial processes (e.g. metal smelting), refineries, shipping, and volcanoes. Within the clean troposphere, the dominant loss of atmospheric SO₂ is oxidation by the hydroxyl radical, resulting in a lifetime of approximately a week. SO₂ contributes to the formation of fine particulate matter that is directly linked with negative health outcomes, particularly
cardiovascular diseases (Khalaf et al., 2024). SO₂ also has broader environmental impacts, primarily by forming sulphuric acid via heterogeous chemistry, which leads to ecosystem damage and building corrosion. The formation of sulfate aerosols affects climate forcing both directly, by scattering incoming sunlight and causing a net cooling of the atmosphere, and indirectly, by perturbing cloud microphysics. In this study, we use machine learning to identify automatically permanent and ephemeral hotspots of SO₂ observed by the TROPOMI satellite instrument and quantify the corresponding emission estimates, carefully





curating sources of error. We will focus on large point sources from fossil fuel combustion, copper smelting, and volcanic emissions.

    Emissions of $SO_2$ from fossil fuel combustion hinges on three factors: the sulfur content of the fuel, combustion efficiency, and the deployment of $SO_2$ scrubber technology. More energy is released during the combustion of coal with a higher number of hydrogen:carbon bonds, which is inversely proportional to the sulfur content. Scrubber technology, widely adopted by

coal-fired power plants in developed nations starting in the 1990s (Srivastava et al., 2001), was introduced to meet regulatory requirements to mitigate acid deposition that caused widespread destruction of downwind forest and aquatic ecosystems (Smith, 1872; Driscoll et al., 2001). Global anthropogenic $SO_2$ emission estimates varied from 93 to 108 Tg yr$^{-1}$ between 2010 and 2018 (Soulie et al., 2023), with recent years showing lower values. While different inventories, such as the Copernicus Atmosphere Monitoring Service (CAMS) and EDGAR, generally agree within $\simeq$4 Tg $SO_2$ yr$^{-1}$ estimates diverge in later

35   years, primarily due to differing estimates for power generation ($\simeq$44% of CAMS anthropogenic emissions during 2010—2018) and shipping ($\simeq$10%). International shipping is a significant source due to the high sulfur content of marine fuel, but low-sulfur fuel regulations introduced in 2020 are expected to have significantly reduced this annual emission, consistent with a large reduction in observed ship tracks in cloud perturbations (Watson-Parris et al., 2022).

    Extracting copper from mineral ores – predominately chalcopyrite ($CuFeS_2$) – releases $SO_2$ to the atmosphere. The smelting

process involves injecting mineral particles and oxygen-enriched air into a furnace that is heated $\simeq$1500 K, resulting in the sulphide minerals reacting with the injected oxygen that eventually produces $SO_2$. Currently, most copper refineries are in China, India, Japan, Russia, and Chile, with only a few smaller-capacity plants in the United States and Germany. Developed countries capture the $SO_2$ waste product but for refineries in developing countries the capture technology may be unaffordable or unavailable.

Volcanoes represent a natural source of $SO_2$ to the atmosphere. They emit $SO_2$ in vast quantities during eruptive and during passive degassing periods. Sulphur species is a minor constituent in volcanic magma, compared with water and carbon dioxide. The production and subsequent emission of $SO_2$ from volcanoes depends on various factors, including the composition and depth of the magma reservoir and the nature of the eruption. Generally, large, explosive (high pressure) volcanic eruptions release more $SO_2$ to the atmosphere than passive degassing periods, which occur due to the movement of sub-surface magma.

Annual volcanic $SO_2$ emission estimates vary. Estimates inferred from satellite data collected between 2005 and 2015 report an annual mean of 23$\pm$2 Tg yr$^{-1}$ for volcanic degassing, with 30% of those sources exhibiting a positive decadal trend (Carn et al., 2017). However, ground-based data collected at 32 volcanoes over the same period report a mean (median) emission rate of $\sim$9.0 (6.8) Tg $SO_2$ yr$^{-1}$ (Carn et al., 2017), and a subset of these ground-based estimates show that some volcanoes degas at a rate too low to be detected by instruments like the Ozone Monitoring Instrument (Arellano et al., 2021). Only within the

past decade has space-borne sensor technology achieved sufficient sensitivity to accurately detect degassing $SO_2$ emissions, providing crucial data that complements information gathered by ground-based networks, such as the Network for Observation of Volcanic and Atmospheric Change.

    Traditional botton-up emission inventories for $SO_2$ (e.g., EDGAR (EDGARv8.1), CEDS (Hoesly et al., 2018)) suffer from critical limitations for modern monitoring, including significant time lags (often years behind real-time) and poor resolu-





tion, providing only mean values over large areas (e.g., 100s of kilometres) and long durations (e.g., monthly). Capturing sporadic emission events or tracking rapid changes in current sources requires the high temporal resolution offered by near real-time satellite observations, especially emerging data from geostationary instruments (e.g. Global Environmental Monitoring Systems (GEMS, Kim et al. (2020)), Tropospheric Emissions: Monitoring of Pollution (TEMPO Chance et al. (2019)) and Sentinel-4 (Bazalgette Courrèges-Lacoste et al. (2017))) that offer continuous monitoring throughout the sunlit day. To

efficiently exploit these massive stream of data, we employ a machine learning model that rapidly highlights $SO_2$ plumes originating from point sources and quickly estimates their emission rates. This capacity for rapid, fine-grained quantification of emission changes is essential for timely intervention and effective atmospheric management. We showcase our approach using data collected by the Tropospheric Monitoring Instrument (TROPOMI) satellite instrument aboard Sentinel-5P.

In the following section, we describe the data used and the development of the model. In section 3 we report our result. We

conclude the study in section 4.

## 2    Data and Methods

### 2.1    TROPOMI SO$_2$

We use Level 2 total column $SO_2$ data retrieved from TROPOMI, a high-resolution UV–Vis–NIR–SWIR spectrometer, onboard the Sentinel-5P satellite, January 2019–December 2024, inclusively. TROPOMI measures the solar radiation backscatter in the

UV range (around 310-330 nm) where $SO_2$ has a distinct absorption feature. Sentinel-5P was launched in October 2017 into a sun-synchronous orbit with a local equatorial overpass time of 13:30. TROPOMI has a swath width of 2600 km divided into 450 across-track pixels, with a pixel resolution of 3.5×5.5 km (across × along track) at nadir for $SO_2$. This sampling strategy results in near-daily global coverage (Veefkind et al., 2012), subject to cloud-free scenes. In this study, we only use pixels with a quality flag > 0.5, as recommended by the TROPOMI Level 2 Product User Manuals (Veefkind et al., 2012).

### 2.2    U-Net Image Detection Model

To automatically detect plumes of $SO_2$ from TROPOMI data, we use a U-Net style fully convolutional network model to perform image segmentation (Ronneberger et al., 2015; Mukhopadhyay et al., 2015). A U-Net model is designed to produce a pixel-by-pixel classification of an image and has been widely used in medical sciences (e.g. tumour detection) as well as in land cover classification in satellite imagery (e.g. Pan et al., 2020; Ulmas and Liiv, 2020; Bokstaller et al., 2021; Filatov and

Yar, 2022). It follows the basic principle of convolving an image over successive layers to reduce the spatial dimensions and extract feature information and then using transposed convolutions to rebuild the image to the original shape, the schematic of the model architecture creates a "U" shape, as shown in Figure 1. This figure shows a schematic of the model architecture used for the plume detection algorithm. We add Gaussian noise to the training images to improve the robustness of the model before passing the image through four downsampling blocks. Each block consists of a double convolutional layer, a max-pooling layer

and a dropout layer set at 20%. These blocks then feed to one more double convolutional layer to extract patterns in the image





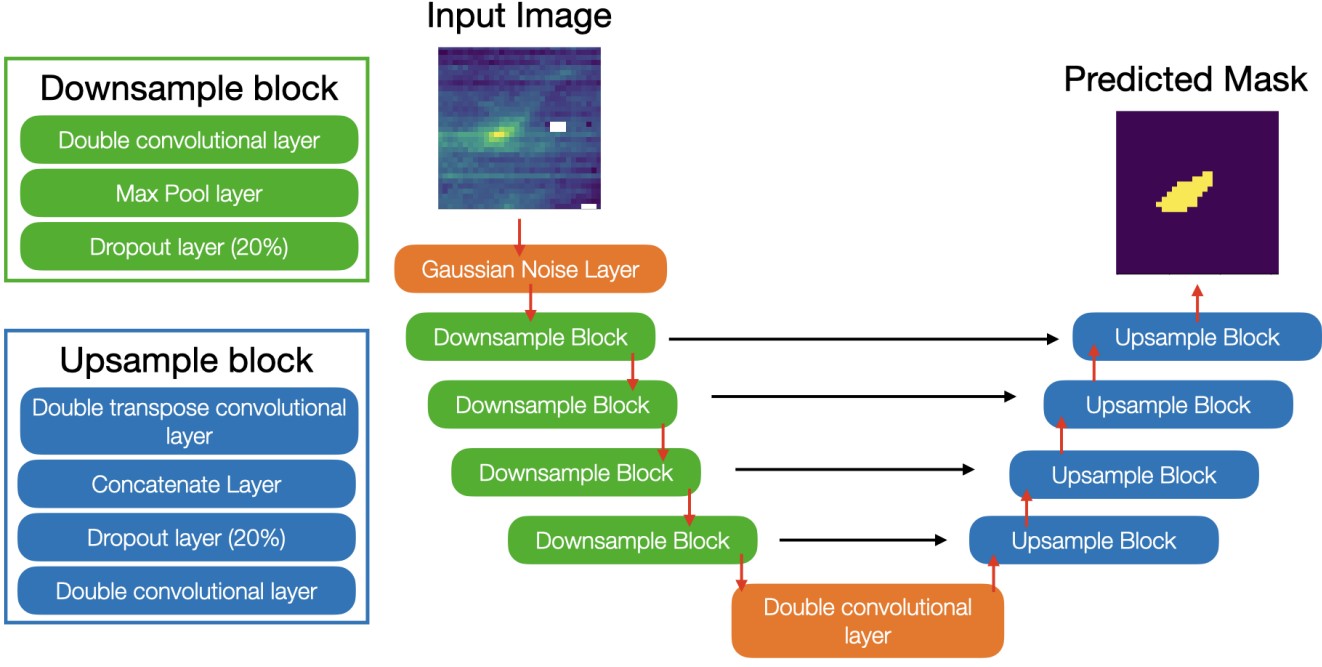

**Figure 1.** Schematic of the architecture of the U-Net model.

before four upsampling blocks create a mask of the plume. Each upsampling block contains a double transpose convolutional layer, a concatenation layer, another dropout layer set at 20% and a double convolutional layer.

Image segmentation models offer significant advantages over traditional image classification (e.g. Finch et al., 2022) because they parse essential information from the background rather than simply assigning a single label to an entire image. For detecting $SO_2$ plumes, this capability is crucial: it allows for more precise geolocating of the plume and the ability to handle multiple distinct plumes within a single satellite scene. Furthermore, segmentation excels over other feature-parsing methods, such as activation maps (Zhou et al., 2015), because it is trained directly on ground-truth image masks. This direct comparison during training provides a clear, quantifiable performance metric, ensuring reliable results.

Our segmentation model was trained using a custom database of over 1,000 TROPOMI images showing $SO_2$ plumes, each paired with a precise plume mask manually created by the lead author. To maximize training effectiveness, we augmented this dataset through rotation and flipping, yielding a final training pool of over 4,000 images and corresponding masks. We chose an image size of 32×32 pixels (roughly 112×176 km² at nadir) as it successfully captures most $SO_2$ plumes. For validation, we trained the model on 80% of the data and tested it on the remaining 20%. The model's performance was measured using precision (correctness of positive predictions) and recall (completeness of positive detection), yielding scores of 65.7% and 74%, respectively. Crucially, the small 32×32 image size disproportionately penalizes minor errors, meaning an offset of just





a few pixels drastically lowers the score. Given this context, we find the model performs adequately to successfully initiate the construction of a comprehensive plume emission database.

To ensure we capture SO$_2$ plumes that may be straddling multiple image boundaries, we employ a 32×32 pixel rolling window that moves four pixels at a time across and along the satellite swath. This systematic sampling generates approximately

110 100,000 images per swath for input into the segmentation model. The model returns each image with a pixel-by-pixel probability of plume presence. We then reconstruct the original swath into an amalgamated mask by taking the median probability of all overlapping pixels. Using the median not only combines the individual predictions but also boosts detection confidence, as an accurately identified plume will appear consistently across multiple overlapping images. We found that a four-pixel step provides adequate coverage to resolve straddling issues while maintaining reasonable computing speed and costs. Crucially,

this overlaying method means the final predicted plume shape is not limited by the 32×32 pixel input size, allowing plumes and their corresponding masks to be accurately mapped across the full scale of the swath (which is typically 4172×450 pixels, spanning 2,600 km from pole to pole).

To extract the details of individual plumes from the reconstructed satellite swath, we apply connected component analysis (using Open-CV (Bradski, 2000)) to the pixel probability array. This analysis effectively identifies the unique plume masks

within the swath and provides the bounding boxes for each one, allowing us to precisely delineate the plume outline. Figure 2 illustrates this capability, showing twenty randomly selected TROPOMI-observed plumes alongside their corresponding predicted plume outlines.

Figure 2 demonstrates that the model generally performs well, although some inaccuracies are present. Detecting atmospheric features like SO$_2$ plumes inherently involves subjectivity, as there is no clear, objective physical boundary for the

125 feature of interest. This human subjectivity is inevitably encoded in the model's training dataset and subsequently reflected in the trained model itself. Continuously refining the model or updating the training dataset is an endless task, so for the practical scope of this paper we have chosen to present results based on a rigorous process involving three training iterations (i.e., checking model output, expanding the dataset with new examples, and retraining).

We have created a comprehensive database documenting each detected plume, which includes its location, date of detection,

plume outline, and an estimate of the emission rate. A bounding box (with a three-pixel buffer) is also recorded, with its limits specifically used to determine the background SO$_2$ concentration outside the plume. Computationally, processing a single swath is highly efficient, taking only about 15 seconds using a GPU or 60 seconds using CPUs.

### 2.3 Source Emission Rates Estimate

By using the predicted plume outline and modelled wind fields, it is possible to calculate an emission estimate associated with

135 a detected plume. To estimate the the emission rate of the source of the plume $E$, we use the following formula:

$$E = \frac{\Delta M \times ws}{L} \times 3600, \qquad (1)$$

where $\Delta M$ is the mass enhancement of the plume relative to the background in kg, $ws$ is median wind speed in ms$^{-1}$ and $L$ is the length of the plume in metres. The results is then multiplied by 3600 to convert from kg s$^{-1}$ to kg hr$^{-1}$.





**Figure 2.** Examples of SO$_2$ plumes in the TROPOMI data and the predicted plume outline, shown as a red line. Warmer colours denote higher values.

We calculate the area of each pixel within the given scene based on the coordinates of the pixel vertices given in the
TROPOMI file which is then used to convert the SO$_2$ observations from mol m$^{-2}$ to grams of SO$_2$ per pixel. The mass
enhancement of the plume is calculated by subtracting the median values of the background, defined as all the pixels within
the bounding box but outside the plume boundary.

To estimate the plume length, we fit an ellipse to the plume outline and use the length of its primary axis, trimming it precisely
to the plume boundary. Figure 3 visually demonstrates this process, showing the mass enhancement relative to the background
median, along with the fitted ellipse and its primary axis. This entire image represents the plume's defined bounding box. As a
specific example, a plume detected over the Cerro Bravo volcano in Colombia (5.13°N, 75.31°W) on December 4, 2018, was
estimated to have a length of 117.4 km and an emission rate of 7071.5 kg hr$^{-1}$.




**Figure 3.** Mass Enhancement (kg) for each plume pixel, shown relative to the background's median value which is defined by the bounding box area outside the plume boundary. The fitted ellipse is displayed in light blue, with its primary axis shown in white. The dashed line indicates where the primary axis was truncated to estimate the plume length. This specific example features an ellipse fit ratio of 0.61.



We use the 10-metre U and V wind fields from ERA5 reanalysis data, included in most TROPOMI Level 2 files, as a foundational estimate for the wind experienced by the emissions. While these near-surface fields may not perfectly represent transport winds - especially for volcanic $SO_2$ injected high into the atmosphere - they provide a reasonable starting point. Although the TROPOMI data includes $SO_2$ layer height, using this to find modelled winds at the precise altitude is computationally expensive and falls outside the scope of this paper.

We quantify how well the ellipse fits the plume shape using the ratio of the plume area to the fitted ellipse area. A ratio closer to 1.0 indicates a better fit, suggesting a more robust emission estimate. This metric highlights plumes with unusual shapes where the primary axis may poorly represent the length. Given the square-pixel nature of the plume shapes, a perfect fit ratio of 1.0 is unrealistic. Based on visual inspection, we consider a good fit ratio to be above 0.4. The example in Figure 3 achieved a ratio of 0.61.

We acknowledge that assumptions inherent in this calculation limit the precision of the emission estimates. However, we believe these estimates serve as a highly useful "first guess" to provide rapid emissions data (sub-one second per plume) from hotspots. The largest source of variability in this calculation is the wind speed variation within the plume. To address this, our plume dataset provides the minimum, maximum, and standard deviation of the wind fields used, allowing users to calculate an approximate range of possible emission fluxes. Furthermore, we include data on the fitted ellipse, plume length, and mass enhancement for users who wish to conduct more thorough investigations.

## 3 Results

### 3.1 Global SO$_2$ Data

We processed approximately 31,000 Level 2 files from TROPOMI observations spanning January 2019 to December 2024. To ensure the reliability of our output, we implemented a filter requiring each detected plume to contain a minimum of six pixels, as the accuracy of smaller predicted plumes is difficult to validate.

In total, 53,993 plumes were identified over the study period. The year 2019 saw the highest number of detections, exceeding 16,000, while the period from 2020 through 2024 showed more consistent annual counts, ranging from ~6,800 to ~8,000 plumes. To locate the emission source, we employ the coordinates of the maximum $SO_2$ concentration within the plume boundary as a practical proxy for the origin, noting that this might not perfectly match the true source location. Figure 4 shows the annual global distribution of these estimated plume origins, 2019–2024.

Initial inspection confirms plumes cluster predictably around known volcanoes and industrial hotspots. However, all years also show regions with a wide, "noisy" spread of detections (e.g., Alaska, Canada, and Eastern Russia annually, plus specific areas like the mid-Atlantic in 2021 or Central Africa in 2024). Figure 5 illustrates the daily plume counts, where each spike corresponds to these noisy geographical spreads (Figure 4). Critically, all but one spike (Peak I in 2023) align with major volcanic eruptions in the region. Table 1 details the specific volcanic emissions believed responsible for this widespread geographical dispersal. Plume transport away from the source is evident, as detections further from the origin often occur in the days immediately following an eruption event (see Appendix A, Figure A1).



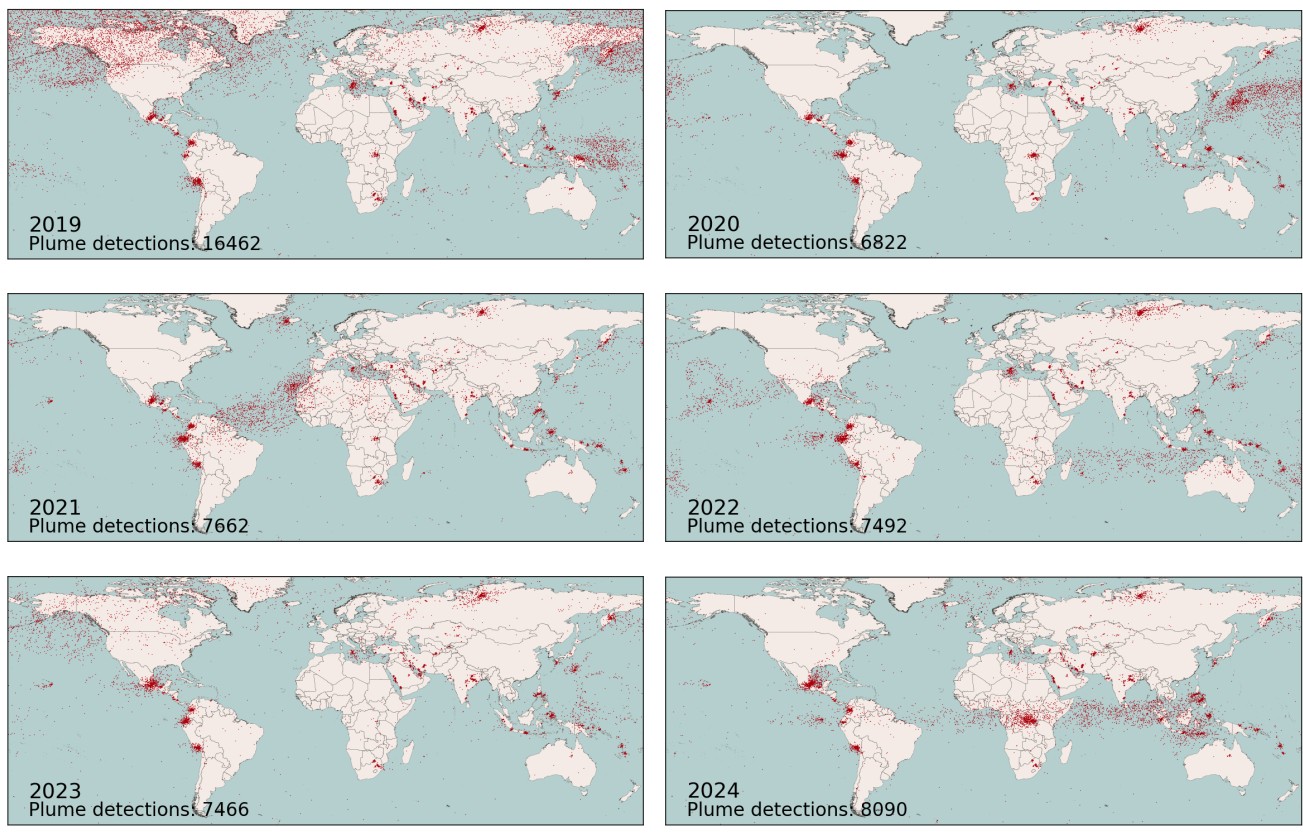

**Figure 4.** Global, annual locations of the maximum concentration of TROPOMI SO$_2$ within the predicted plume masks, 2019–2024.

We attribute the major spike in 2019 plume detections mostly to the coinciding eruptions of Raikoke, Russia, and Ulawun, Papua New Guinea. The Raikoke eruption on June 21st is particularly notable, injecting one of the largest amounts of sulfur dioxide (SO$_2$) into the stratosphere since the 1991 Mount Pinatubo eruption (Vernier et al., 2024; **?**).

While Peak I in Figure 5 does not correspond to a volcanic eruption, the surge in detections is concentrated over Norilsk in Northern Russia (88.1°W, 69.3°N), as shown in Figure 6. This region is a known major source of SO$_2$ emissions due to its large metal smelting operations. The precise cause of this specific increase in plume detections, however, remains unknown.

Using the DBSCAN (Density-Based Spatial Clustering of Applications with Noise) algorithm Ester et al., we grouped the detected plumes based on the coordinates of their maximum SO$_2$ observation (our proxy for origin). We set the clustering parameters to a maximum distance of 50 km between points and a minimum of 20 samples per cluster. This technique effectively identifies global areas with a high concentration of plumes. Figure 7 shows the centers of the 90 detected clusters, coloured





**Figure 5.** Number of plume detections per day for each year of the study. Highlighted regions show time periods of a higher than usual number of detections. Labels correspond to Table 1.





| Label | Vocano Name | Country | Coordinates | First Detection Date |
|---|---|---|---|---|
| A | Raikoke | Russia | 153.25°E, 48.29°N | 22nd June 2019 |
| B | Ulawun | Papua New Guinea | 151.33°E, 5.05°S | 26th June 2019 |
| C | Ulawun | Papua New Guinea | 151.33°E, 5.05°N | 3rd August 2019 |
| D | Taal | Philippines | 120.99°E, 14.01°N | 12th January 2020 |
| E | Mount Cleveland | United States of America | 169.94°W, 52.82°N | 18th June 2020 |
| F | La Soufrière | Saint Vincent and the Grenadines | 61.18°W,13.33°N | 11th April 2021 |
| G | Hunga Tonga-Hunga Ha'apai | Tonga | 175.38°W, 20.55°S | 14th January 2022 |
| H | Mauna Loa | United States of America | 155.60°W, 19.475°N | 26th November 2022 |
| I | - | - | - | - |
| J | Taal | Philippines | 120.99°E, 14.01°N | 14th April 2024 |

**Table 1.** Major volcanic eruptions thought to be responsible for the periods of higher than usual plume detections highlighted and labelled in Figure 5.

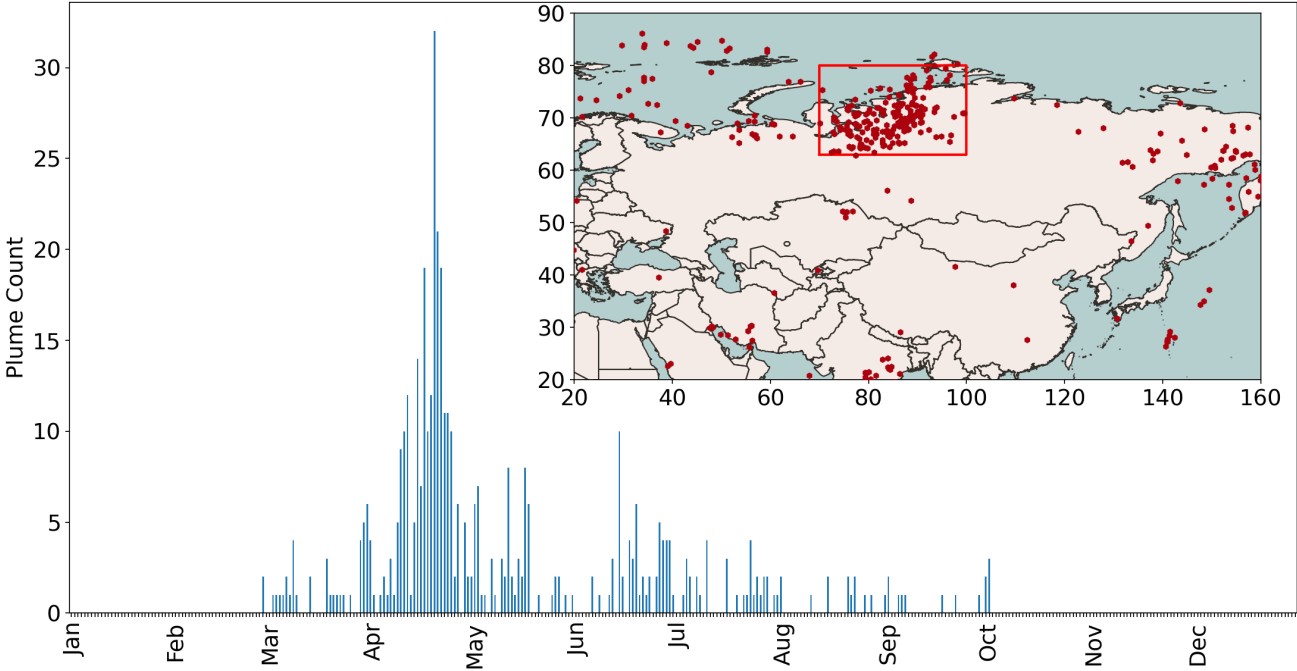

**Figure 6.** Number of plume detections per day for during 2023 within the red box on the inset map (centred over Norilsk, Russia).



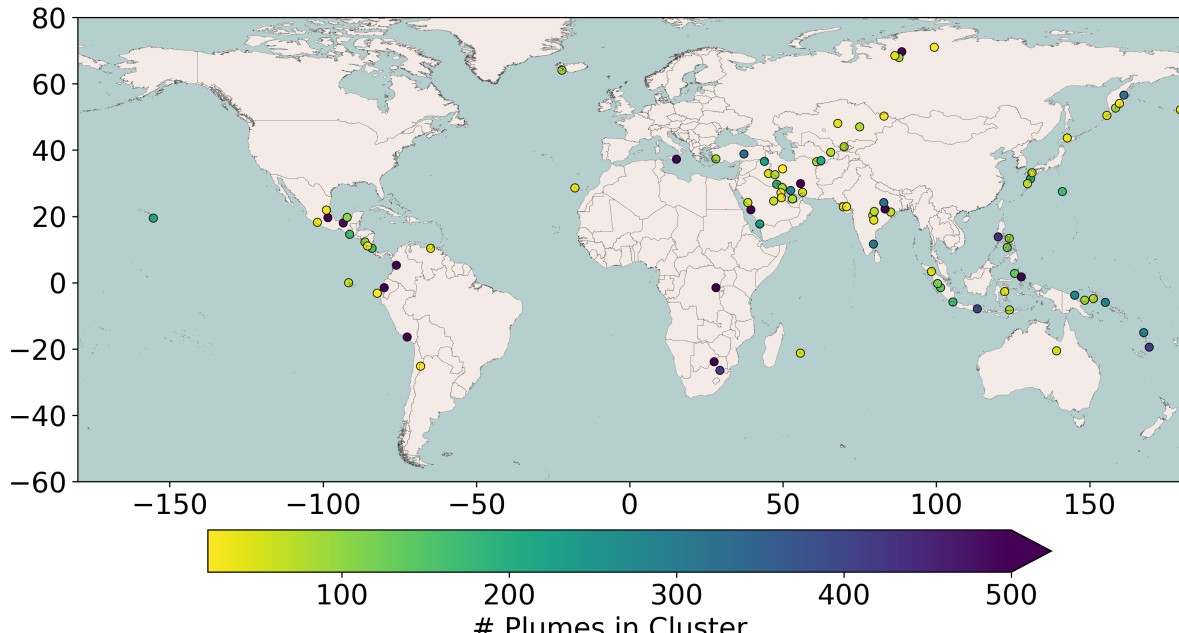

**Figure 7.** Locations of the centre of a cluster of plumes coloured by number of plumes in that cluster.

by the number of plumes they contain, which further validates that our model successfully identifies the major SO$_2$ sources worldwide.

We observed regions on the global maps, most notably China (a location with numerous known SO$_2$ sources), where the model failed to detect expected plumes. Assuming the quality and volume of TROPOMI data are consistent globally, this

deficiency likely stems from the plume detection model itself. We hypothesize that high background SO$_2$ concentrations in such regions may "hide" point sources, making plumes difficult to isolate. Since the model was trained on global data, it may struggle with these outlier scenarios. Future iterations should address this by specifically training on data that includes plumes from high-background regions like China.

Table 2 details the ten largest clusters of SO$_2$ plumes detected over the six-year study, alongside their probable sources.

Volcanic activity dominates this list, accounting for seven out of the ten largest clusters. The remaining three clusters are associated with industrial activities: metal smelting, coal mining, and oil and gas operations. It is crucial to note that these clusters are often complex; some may contain multiple source types. For example, the Iztaccíhuatl volcano cluster is located just south of Mexico City, meaning the cluster likely includes industrial SO$_2$ sources alongside the volcanic emissions.

### 3.2    Coincidence with Volcanic Activity

Using the Smithsonian Institution's Global Volcanism Program (GVP) database of known eruptions since 1960 (GVP), we identified 227 active global eruptions during our study period. Comparing this list with our plume database revealed that 7,943



| Rank | Cluster Centre Coordinates | Nearest Likely Source | Source Type | # of Plumes | Median Emission Rate (kg hr$^{-1}$) | Daily Persistence (%) |
|---|---|---|---|---|---|---|
| 1 | 19.45°N, 98.54°W | Iztaccíhuatl, Mexico | Volcano | 1830 | 15,277 | 55.4 |
| 2 | 16.27°S, 72.17°W | Ampato, Peru | Volcano | 1628 | 25,859 | 49.9 |
| 3 | 1.50°S, 80.01°W | Chimborazo, Equador | Volcano | 1604 | 8582 | 42.7 |
| 4 | 5.26°N, 76.08°W | Nevado del Ruiz, Columbia | Volcano | 1035 | 9716 | 37.2 |
| 5 | 69.65°N, 88.74°E | Norlisk, Russia | Metal Smelting | 875 | 12,252 | 25.0 |
| 6 | 1.75°N, 127.71°E | Dukono, Indonesia | Volcano | 782 | 9268 | 30.2 |
| 7 | 1.55°S, 28.17°E | Mount Nyiragongo, DRC | Volcano | 668 | 7439 | 24.5 |
| 8 | 23.74°S, 27.43°E | Grootegeluk Mine, S. Africa | Coal Mine & Power | 612 | 12,769 | 26.0 |
| 9 | 37.24°N, 15.25°E | Mount Etna, Italy | Volcano | 490 | 23,447 | 20.1 |
| 10 | 21.99°N, 39.51°E | Rabigh, Saudi Arabia | Oil and Gas | 520 | 15,005 | 19.4 |

**Table 2.** Top ten largest clusters of plumes detected and their probable source, the total number of plumes over the six year study period, the median emission rate for all plumes in the cluster and the percentage of days a plume is detected in this region.

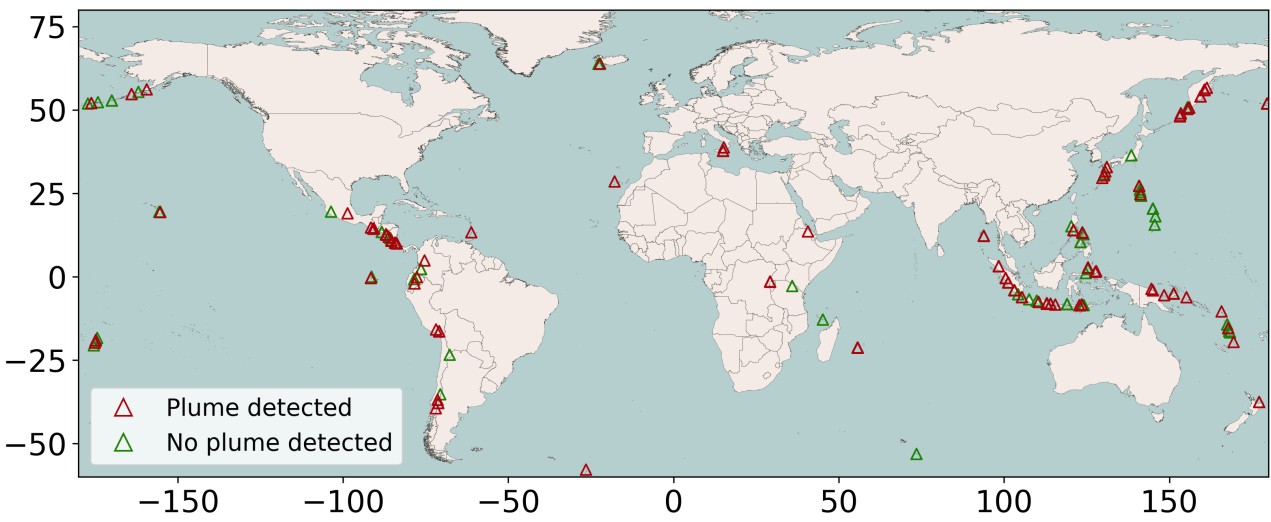

**Figure 8.** All eruptions reported between January 2019 and December 2024. Red triangles indicate plumes were detected within 50 km radius of the eruption during the eruptions dates. Green indicate no plume was detected.

plumes were detected within a 50 km radius of an eruption during its active date. This confirms plume detection for 110 (49%) of these eruptions. The 117 missing detections are likely due to poor TROPOMI retrievals caused by high cloud cover or heavy aerosol loading. Figure 8 maps these GVP-reported eruptions, coloured by whether plumes were detected nearby (red) or not
(green).



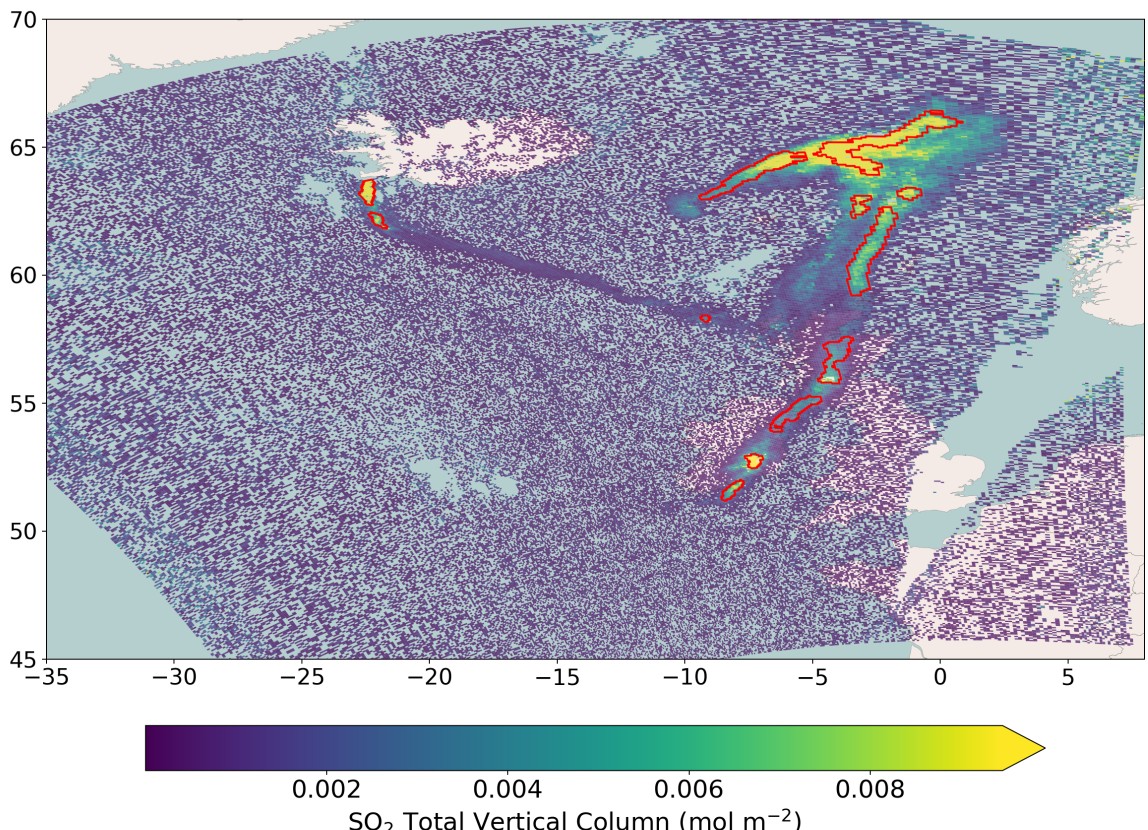

**Figure 9.** TROPOMI SO$_2$ total vertical column over the United Kingdom on 8th August 2024 showing a plume from the Sundhnúkur volcano in Iceland.

The plume detection algorithm struggles with very large plumes (> 1000 km) that often occur far downwind of major eruptions. Figure 9 illustrates this, showing a large SO$_2$ plume over the United Kingdom on August 8, 2024, originating from the Sundhnúkur volcano in Iceland (63.8°N, 22.38°W) six days earlier. The red outlines show the model splits this large plume into multiple smaller segments. This error likely results from the method used to break the TROPOMI swath into smaller images, which prevents the model from seeing the larger context of a plume that can span the entire swath.

### 3.3 TROPOMI SO$_2$ Activity Detection Flag

The TROPOMI Level 2 data assigns a flag to SO$_2$ pixels based on five categories: (0) no enhancement, (1) general SO$_2$ detection, (2) near a known volcano, (3) near a known anthropogenic source, and (4) a potential false positive due to a high solar zenith angle. Since a single plume often covers many pixels, we assign the final plume label only if over 80% of its pixels share the same TROPOMI flag (likely determined by spatial proximity to known sources); otherwise, the plume is





**Figure 10.** Global distribution of plume labels, 2019 - 2024, linked with the TROPOMI flag assigned to Level 2 SO$_2$ data.

labeled as a combined source. We found that the dominant category is a general SO$_2$ detection with no source attribution (43.5% of plumes), followed by volcanic plumes (30.8%) and known anthropogenic sources (14.7%). Figure 10 illustrates the geographical distribution of these categorized plumes.

The high number of unlabelled SO$_2$ detections is likely caused by plumes that have traveled downwind from their origin 225 and are thus no longer close to the known volcanic or anthropogenic sources used for TROPOMI flagging. This explanation is strongly supported by the evidence of extensive volcanic outflow visible in both the geographical distribution of plumes (Figure 4) and the daily detection spikes (Figure 5).

The TROPOMI "No enhanced SO$_2$ detected" category accounts for 10.0% of all detections, but we believe many of these should be attributed to an actual source rather than being false positives. We demonstrate this by applying the clustering 230 algorithm (as described in Section 3.1) to these "no enhanced detection" plumes. This allows us to extract meaningful groupings and calculate their proximity to clusters with known sources. Figure 11 displays all plumes in this category, highlighting 34



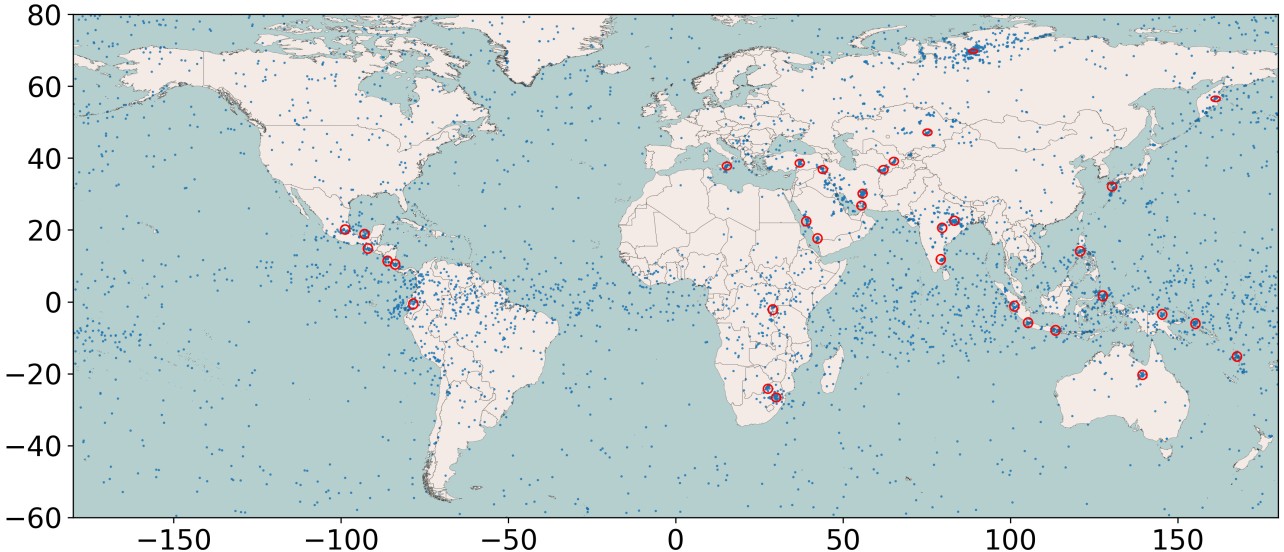

**Figure 11.** Location of all plumes labelled as "no enhanced detection" (blue), along with 34 clusters of more than 20 plumes within 150 km of each other (red) circles.

dense clusters (minimum 20 plumes within 150 km). Crucially, 27 of these 34 clusters overlap with clusters that have a known emission source, and the remaining seven are within 300 km. While these clusters account for 24% of the "no enhanced detection" plumes, many of the remaining plumes form a large, dispersed swath extending from 100°E to 180 °W around

10°S. This pattern aligns strongly with the downwind transport of $SO_2$ from the Taal Volcano eruption in the Philippines (120.99°E, 14.01°N) in April 2024 (detailed in Section 3.1).

This analysis demonstrates that while the source labelling provided in the TROPOMI files is informative, it fails to capture the complete picture of emission attribution. Although our plume detection algorithm does not attribute sources itself, it successfully highlights geographical and temporal patterns that are crucial for identifying important, and potentially unlabelled,

$SO_2$ emission locations.

## 3.4   Source Emission Rate Estimates

Using the methodology detailed in Section 2.3, we calculated the emission rate for every detected plume. Figure 12 provides histograms illustrating the distribution of these emission rates, plume lengths, the ellipse fit metric, and the coefficient of variation (CoV) of the wind fields across the entire study period. The overall median emission rate for all plumes is 14,629

245   kg hr$^{-1}$ (Figure 12A). These rates assume the source originates within a single TROPOMI pixel. By examining the lower



percentiles to account for data noise, we estimate a practical detection limit for our plume detection algorithm to be the 1st percentile of emission estimates, which corresponds to 524 kg hr$^{-1}$.

Figure 12B shows that the distribution of plume lengths is strongly skewed toward shorter plumes, with a median length of 37.3 km. This confirms that most plumes are relatively small compared to the 2600 km TROPOMI swath, often spanning only about 10 pixels at nadir. The histogram for the ellipse fit metric (Figure 12C) has a median value of 0.6, which validates the assumption (discussed in Section 2.3) that fitting an ellipse to determine plume length holds true in the majority of cases. Finally, the wind field's CoV (Figure 12D), calculated as $\bar{ws}/\sigma ws \times 100$, quantifies wind speed variability within each plume. CoV values close to 0% support the use of the median wind speed as representative. The plume dataset shows a median CoV of 12.5%, indicating that most plumes experience only minor variations in wind speed.

All these metrics are available for individual plumes, and therefore can be used to help filter results for specific use cases.

## 3.5 Comparison with EDGAR Emission Dataset

Comparing our plume emission database to existing SO$_2$ emission datasets like EDGAR is non-trivial and must be interpreted cautiously, as the datasets serve fundamentally different purposes. A direct grid-to-grid comparison is uninformative because our algorithm only captures emissions above a certain threshold and requires clean TROPOMI observations, resulting in an incomplete picture of global emissions. Conversely, gridded inventories often fail to capture extreme or sporadic emission events. Furthermore, our detection algorithm does not distinguish between anthropogenic and volcanic sources, which complicates direct comparisons. Figure 13 shows the cumulative emission profile from EDGARv8.1 for 2022 and highlights the issue: global SO$_2$ emissions are heavily skewed toward a few large point sources. For instance, the largest 1% of sources contribute 85% of total global emissions, and the top 0.1% account for 50%. This inherent heterogeneity means the majority of grid squares in the EDGAR data contain fluxes below the detection threshold of our algorithm.

We aggregate the detected plumes onto a monthly $0.1° \times 0.1°$ regular grid to align with the EDGAR emission database and compare flux estimates where plumes were found. Figure 14 shows that the SO$_2$ flux estimated from our plume detections is roughly four orders of magnitude larger than the typical fluxes reported in the EDGAR database for the same grid squares. As detailed in Section 3.4, our assumed detection limit of 524 kg hr$^{-1}$ (which equates to $3.81 \times 10^{-9}$ kg m$^{-2}s^{-1}$ on the EDGAR grid) surpasses 99.8% of EDGAR's reported emissions. This stark difference underscores the value of the plume database as a specialized tool for detecting irregular, high-intensity emission events rather than serving as a complete, comprehensive emission inventory.

To assess how well our plume database represents persistent, high-emission areas, we compare the location of detected plumes against the largest 0.1% of EDGAR emission sources. The results show poor agreement: our algorithm detects a plume within 200 km of these major sources only 9.4% of the time. Figure 15 highlights the locations of EDGAR emissions above this 0.1% threshold where no plume was detected. The map clearly indicates a significant number of these undetected high-emission regions clustered over China and India. As noted previously (Section 3.1), the high density of EDGAR's large emission fluxes in eastern China strongly suggests that our plume detection algorithm is unable to isolate individual plumes in regions where the background SO$_2$ concentration is excessively high.



**Figure 12.** Histograms of the (A) estimated emission rates, (B) plume length, (C) ellipse fit metric, and (D) coefficient of variation for wind speed for all plumes detected during the study period.





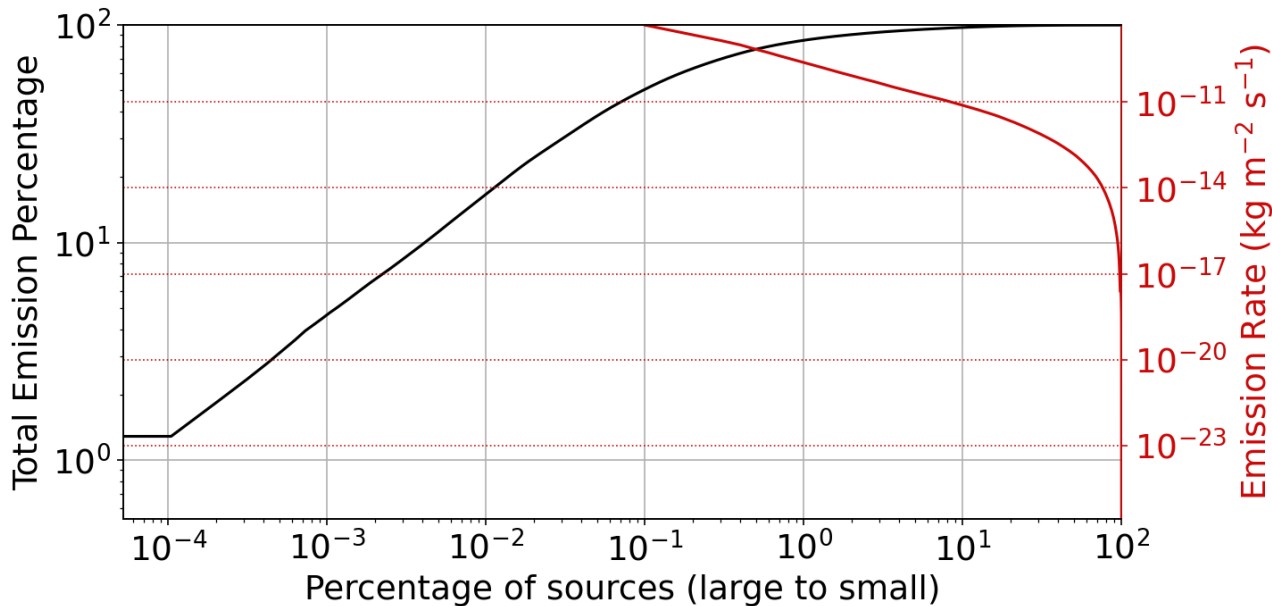

**Figure 13.** The cumulative percentage of total emissions (black) and the corresponding emission rate (red) as a function of percentage of sources, from large to small

## 4  Concluding Remarks

The segmentation model successfully processed approximately 31,000 TROPOMI Level 2 files from 2019 to 2024, demonstrating scalability and efficiency, with a rapid processing time of about 15 seconds per swath on a GPU. The methodology, which employed a $32 \times 32$ pixel rolling window and median probability overlap, effectively addressed issues related to plumes straddling image boundaries while ensuring the final plume shape was not constrained by the input image size. Although the raw performance metrics (Precision: $65.7\%$, Recall: $74\%$) may appear modest, they are negatively skewed by the high sensitivity of using small $32 \times 32$ pixel images for validation. The model was deemed sufficient to establish a viable "first guess" emission database.

Our assumptions inherent to the emission rate calculation method (Section 2.3) were statistically validated by the resulting dataset. The median ellipse fit metric of 0.6 confirms that fitting an ellipse to determine plume length holds true in the majority of cases. Furthermore, the wind field analysis yielded a low median Coefficient of Variation (CoV) of just 12.5%, supporting the assumption that the median wind speed accurately represents transport conditions within most plumes. Based on the lowest detected emissions, we conservatively inferred a practical detection limit for the algorithm of 266 kg hr$^{-1}$.

The global analysis confirmed that plumes cluster predictably around known volcanic and industrial hotspots. The detection of 53,993 plumes highlighted the critical role of episodic volcanic activity in SO$_2$ budgets, with significant annual spikes





**Figure 14.** Histogram of the emission rates from EDGAR where plumes are detected (green) and the gridded plume emission estimates (red)





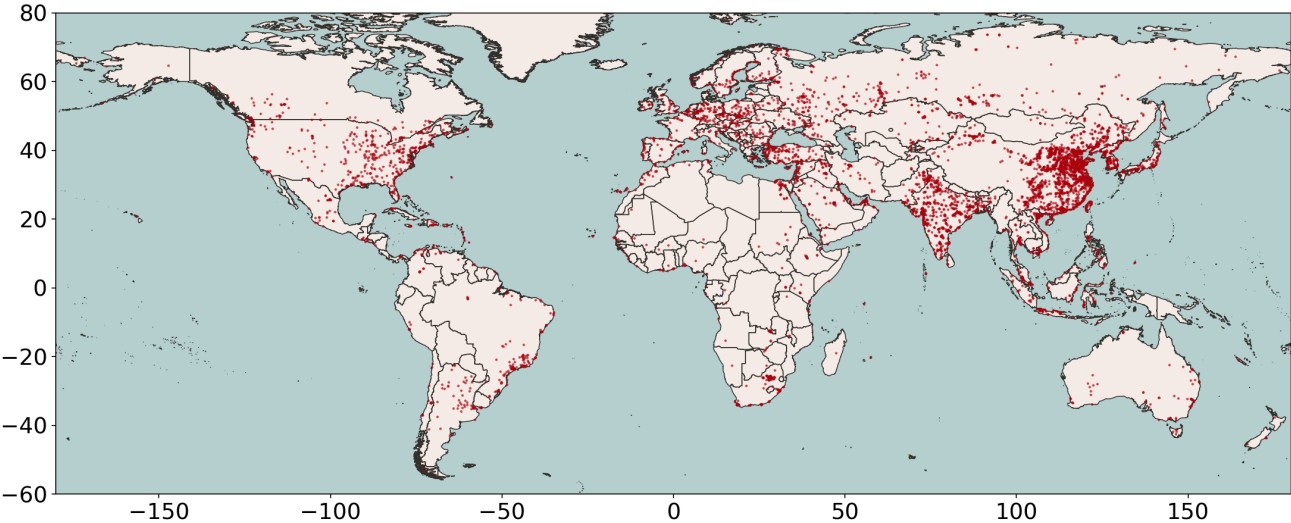

**Figure 15.** Locations of 2022 EDGAR emissions above $4.8 \times 10^{-9}$ kg m$^{-2}$ s$^{-1}$ where no plume was detected within 200 km.

attributed to major events like the Raikoke and Ulawun eruptions in 2019. We also found that the TROPOMI Level 2 source
flagging is incomplete; 43.5% of detections were unlabelled, a finding we attribute to plumes being transported far downwind
from their designated source areas. Clustering analysis of the "No enhanced SO$_2$ detected" plumes (10.0% of detections)
showed that many are, in fact, real sources, often corresponding to large, diluted outflow features like the plume from the 2024
Taal eruption.

However, two primary limitations must be addressed in future work. First, the algorithm struggles with very large plumes
($> \sim$ 1000 km), often incorrectly segmenting them into multiple smaller plumes due to the inherent context limitation of the
small image input method. Second, and more critically, there is a pronounced lack of plume detection over expected high-
emission industrial regions like China and India. The poor spatial agreement with EDGAR's largest 0.1% sources (detecting a
plume only 9.4% of the time) strongly suggests that the algorithm fails to isolate individual plumes against an excessively high
SO$_2$ background concentration prevalent in these regions.

Finally, the comparison with the gridded EDGAR emission database highlighted the distinct utility of our dataset. Our
detected fluxes were found to be approximately four orders of magnitude larger than typical EDGAR fluxes in the same grid
cells. This disparity confirms that our database is not intended as a complete emission inventory; rather, it is a specialized
tool designed to efficiently and rapidly capture and quantify high-magnitude, transient, and irregular emission events—such as
major volcanic eruptions and large sporadic industrial spikes—that are typically smoothed out or omitted by standard annual
inventories.





A crucial area for future advancement lies in integrating data from recently launched geostationary satellites. While the TROPOMI analysis relies on a single daily snapshot, instruments like the Geostationary Environment Monitoring Spectrometer (GEMS, covering Asia), the Tropospheric Emissions: Monitoring of Pollution (TEMPO, covering North America), and the future Sentinel-4 (covering Europe) offer sub-hourly or hourly measurements. This significantly higher temporal resolution would be transformative for this work, allowing for more robust plume tracking and improved wind field context for emission estimation. Furthermore, the hourly data would greatly enhance our ability to differentiate transient, high-flux plume events from persistent, high background $SO_2$ concentrations, thereby providing a pathway to potentially overcome the detection issues currently observed in heavily polluted industrial regions like China.

The application of machine learning is essential for achieving near real-time $SO_2$ source analysis, particularly for rapidly evolving natural hazards like volcanic eruptions . The immense processing speed of the segmentation model (sub-second per plume) is crucial for aviation safety, as $SO_2$ serves as a key proxy for hazardous volcanic ash. Machine learning automates the complex, time-consuming steps of plume boundary definition, geometric fitting (e.g., length), and emission rate calculation, transforming raw TROPOMI data into quantitative, actionable intelligence instantaneously. This rapid, objective assessment informs Volcanic Ash Advisory Centers and provides timely input for atmospheric transport models, greatly enhancing warning systems and safety protocols.

*Code availability.* The plume detection code can be requested from the authors.

*Data availability.* The $SO_2$ plume detection dataset can be requested from the authors. TROPOMI $SO_2$ data are available from https://dataspace.copernicus.eu/

**Appendix A: Major Volcanic Outflow**

Figure A1 shows the plumes detected near likely volcanic eruptions, coloured by the days elapsed since the first eruption was detected. These show how these plumes are likely outflow from these large eruptions which are likely being detected on subsequent days as they are transported downwind.

**Appendix B: Plume Dataset Description**

The resulting plume dataset described in this paper is available for use and contains the following information:

- Date and time of the TROPOMI swath

- Latitude and longitude of the maximum $SO_2$ value within the plume (degrees north and west)

- X and Y Index on the TROPOMI swath of location of the maximum $SO_2$ value within the plume





**Figure A1.** Plumes likely associated with major volcanic eruptions through the study period. The colouring represents days elapsed since first eruption detected.





- Value of the maximum $SO_2$ concentration within the plume (mol m$^{-2}$)

- Indices on the TROPOMI swath of the bounding box surrounding the plume

- Plume mass enhancement from the background (kg)

- Latitude and longitudes of the plume border (degree north and west)

- Indices on the TROPOMI swath of the plume border

- Data on the fitted ellipse (x and y of ellipse centre, width, height and angle)

- Median, minimum, maximum, standard deviation and standard error of wind speed (ms$^{-1}$)

- Wind direction (degrees)

- Plume length (m)

- Plume emission estimate (kg hr$^{-1}$)

- Array of pixel areas within the bounding box (m$^2$)

- U and V wind fields within the bounding box

- Name of the original TROPOMI file used

*Author contributions.* DPF and PIP designed the research; DPF prepared the calculations; DPF and PIP analysed the results and wrote the paper.

*Competing interests.* There are no competing interests present.

*Acknowledgements.* Douglas P. Finch and Paul I. Palmer gratefully acknowledge funding from the ESA.



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
