# Peer review of "A machine-learning reference dataset for SO2 plumes observed by TROPOMI: uncertainties and emission estimates"

_EGUsphere, 2025_

## Community Comment (CC1)

**Comment on egusphere-2025-5900**

The manuscript „A machine-learning reference dataset for SO2 plumes observed by TROPOMI: uncertainties and emission estimates" from Dougls P. Finch and Paul I. Palmer presents a machine learning approach to detect SO2 plumes in TROPOMI SO2 data using a U-Net image segmentation model.

In my view, the title of the manuscript is somewhat misleading. While it suggests that a "reference dataset" is provided alongside the paper as supplementary material, the dataset must instead be requested directly from the authors. Therefore, the dataset should be provided with the paper or the title should be changed. Moreover, it remains unclear, why this dataset should be considered a "reference", given that the underlying method to detect SO2 pixels lacks clarity in several aspects including the error characterization. It is also not evident how the proposed method is an improvement over the existing SO2 detection flag included in the operational TROPOMI products. The authors state, that the model was trained "with a precise plume mask manually created by the lead author" but provide no further details (see discussion below). In my opinion, the training of the algorithm could be substantially improved by incorporating the operational TROPOMI SO2 detection flag into the training samples.

Although the authors acknowledge funding from ESA, the manuscript does not adequately reference the relevant publications and documentation associated with the ESA TROPOMI SO2 product and its characteristics.

Overall, I believe the paper would benefit from restructuring, rewriting, and improvements to the training approach prior to acceptance, as detailed in the following comments.

**Detailed comments:**

Introduction:

This section should already mention the SO2 detection flag in the operational TROPOMI product (this appears only in the Results section 3.3) and should discuss why a image segmentation model could potentially be better in detecting SO2 plumes

Section 2.1

Although the authors correctly summarize the TROPOMI SO2 L2 product overall, they do not follow standard practices and cite the related publications: Theys et al. 2017) and the TROPOMI SO2 Algorithm Theoretical Baseline Description (ATBD). This section should also describe the SO2 detection algorithm in detail, so the authors can refer to it later in their results and discussions.

Section 2.2

Lines 99-100: The authors write that they train their model with a "custom database ... with a precise plume mask manually created by the lead author." Please add details how the database was generated, how the plume mask was generated, including SO2 thresholds, how many samples are used for the training, which time period was covered, etc.

Is there a reason, why the authors did not use the operational TROPOMI SO2 detection flag for generating training samples? This would improve the algorithm significantly, especially for weak plumes as well as extended plumes, which the algorithm is currently struggling with.

An error characterization of the image detection model should be included in this section.

Figure 2: Please add a colorbar with SO2 VCD values. Does the figure only show 32x32 pixel subimages which contain a detected plume? Consider showing the actual pixel-wise detection mask.

Section 2.3

Line 140: How was the mass enhancement calculated? I assume you used the *sulfurdioxide_total_vertical_column* variable in the main PRODUCT group of the TROPOMI SO2 L2 files... Please note that this VCD is representative for a SO2 pollution source close to the ground. Actual volcanic SO2 VCDs for assume plume heights of 1,7,15km can be found in the DETAILED_RESULTS subgroup of the L2 product and should be used to calculate mass enhancements of volcanic plumes.

Lines 148-152. It is correct that the 10m U and V wind fields only serve as a first estimate, but when you analyze volcanic plumes ,the 10m wind fields are not applicable at all. Here the SO2 layer height in the product (please refer to Hedelt et al. 2019 and Koukouli et al 2021 for details) can be used to interpolate in ERA-5 altitude-resolved wind field data.

Section 3.1

I am missing a clear statement in this section of what is the lowest SO2 amount the algorithm can detect here.

Lines 184ff: From daily TROPOMI SO2 images and news entries it is clear that the Shiveluch volcanic eruption produced the huge SO2 plume in period "I", which was detected by the algorithm.

Figure 9: The map shown is not from 08 August 2024 but from 24 August 2024. Attached is a map of pixels identified by the operational SO2 detection flag of TROPOMI :

[Figure]

It is clear that the operational TROPOMI flag identifies many more pixels as enhanced SO2 compared to the image classification. Therefore, I would suggest that the authors train their model using the operational detection flag to improve their results.

Section 3.3.

Line 216ff: As described before, the TROPOMI flag should be introduced much earlier in the manuscript together with the corresponding references.

Figure 11. The distribution of "no enhanced detection" shows many false positive detection scattered around the globe with an enhancement over the mid-latitudes (high cloud cover) as well as many detections at high latitudes (either high SZA and/or high albedo from snow cover). I suggest to refine the model and restrict it to low SZA below about 60-65 degrees, and also take into account additional information in the training (e.g. SZA, cloud cover, surface /cloud albedo)

Line 237ff "This analysis demonstrates that while the source labelling provided in the TROPOMI files is informative, it fails to capture the complete picture of emission attribution..." This summary is very questionable and misleading. The operational TROPOMI SO2 flag is flagging pixels with enhanced SO2, with a low threshold of >0.2DU and a distance-based labelling based on known volcanic and anthropogenic locations. It is therefore not at all designed for emission attribution. The same applies to the new plume detection algorithm presented by the authors. The new algorithm does not even indicate the potential source (anthropogenic or volcanic). So therefore the whole statement should be carefully rewritten.

MISSING REFERENCES

Theys et al. 2017: https://doi.org/10.5194/amt-10-119-2017

TROPOMI SO2 L2 specifications, dataset, ATBD: *https://doi.org/10.5270/S5P-74eidii*

Hedelt et al. 2019 https://doi.org/10.5194/amt-12-5503-2019

Koukouli et al 2021 https://acp.copernicus.org/articles/22/5665/2022/acp-22-5665-2022-discussion.html